# Assessment of the Effects of Chitosan, Chitooligosaccharides and Their Derivatives on *Lemna minor*

**DOI:** 10.3390/molecules27186123

**Published:** 2022-09-19

**Authors:** Bianca-Vanesa Boros, Daniela Dascalu, Vasile Ostafe, Adriana Isvoran

**Affiliations:** 1Department of Biology-Chemistry, Faculty of Chemistry, Biology, Geography, West University of Timisoara, 16 Pestalozzi, 300115 Timisoara, Romania; 2Advanced Environmental Research Laboratories (AERL), 4 Oituz, 300086 Timisoara, Romania

**Keywords:** chitosan, chitooligosaccharides, *Lemna minor*, ecotoxicity, aqueous environment

## Abstract

Chitosan, chitooligosaccharides and their derivatives’ production and use in many fields may result in their release to the environment, possibly affecting aquatic organisms. Both an experimental and a computational approach were considered for evaluating the effects of these compounds on *Lemna minor*. Based on the determined EC_50_ values against *L. minor*, only D-glucosamine hydrochloride (EC_50_ = 11.55 mg/L) was considered as “slightly toxic” for aquatic environments, while all the other investigated compounds, having EC_50_ > 100 mg/L, were considered as “practically non-toxic”. The results obtained in the experimental approach were in good agreement with the predictions obtained using the admetSAR2.0 computational tool, revealing that the investigated compounds were not considered toxic for crustacean, fish and *Tetrahymena pyriformis* aquatic microorganisms. The ADMETLab2.0 computational tool predicted the values of IGC_50_ for *Tetrahymena pyriformis* and the LC_50_ for fathead minnow and *Daphnia magna*, with the lowest values of these parameters being revealed by totally acetylated chitooligosaccharides in correlation with their lowest solubility. The effects of the chitooligosaccharides and chitosan on *L. minor* decreased with increased molecular weight, increased with the degree of deacetylation and were reliant on acetylation patterns. Furthermore, the solubility mainly influenced the effects on the aqueous environment, with a higher solubility conducted to lower toxicity.

## 1. Introduction

Chitosan is a linear polysaccharide obtained by the deacetylation of chitin, a polymer widely found in nature. While chitin contains especially N-acetyl-D-glucosamine units, chitosan consists predominantly of D-glucosamine units, with randomly acetylated D-glucosamine units also being present. Due to the presence of the hydroxyl (-OH) and amino (–NH_2_) groups, chitosan is easily available for chemical reactions, having many special properties. These properties are influenced by both the degree of deacetylation (DaD) and the molecular weight (MW), the biological effects and the water solubility enhanced by higher DaD and lower MW of chitosan [1,2].

The remarkable macromolecular structure, properties and biological effects of chitosan give it a wide range of applications in many fields: medicine, pharmacy, the food industry, agriculture, cosmetics, packaging, nanotechnology, etc. [3,4,5,6,7,8,9]. The wide range of applications of chitosan in the various forms used can lead to a possible contamination of the environment with this polysaccharide, either as waste or as accidental spills. Thus, the identification of the potential ecotoxicological effects of chitosan is imperative.

There is a reduced number of studies that assess the ecotoxicological effects of chitosan, with a review being published in [10]. The synthesis of the available data shows that chitosan exhibits both morphological and biochemical effects towards fungi. For chitosan with high DaD (60–95%), biochemical and genetics effects were observed in plants, while both morphological and biochemical effects were observed in worms [10]. A high insecticidal effect of chitosan with DaD = 80% and MW = 300 kDa was observed for lepidopterous and homopterous insect pests [11]. Chitosan with DaD between 70 and 75% showed various degrees of inhibition on several pathogen microorganisms [12]. The growth of algae such as *Chlorella vulgaris* was inhibited by exposure to chitosan with DaD of 90% [13].

Chitosan derivatives with improved properties can be obtained by chemical modifications of chitosan by enlarging the main chitosan application fields [14,15,16,17], which can also lead to a potential environmental risk. Carboxymethyl chitosan (CMChi), one of the derivatives of chitosan, is obtained by the carboxymethylation of chitosan, which can occur at hydroxyl group, at amino group, or at both, generating three CMChi types: O-carboxymethyl chitosan (O-CMChi), N-carboxymethyl chitosan (N-CMChi) or N,O-carboxymethyl chitosan (NO-CMChi). CMChi has numerous applications in diverse fields [18,19]. Literature data reveal that carboxymethyl chitosan had antimicrobial effects against some bacterial strains [12] or caused an abnormal growth of fungi [20].

Chitooligosaccharides (COs) are obtained by the depolymerization of chitosan [21]. The name of COs is common for chitosan products having low degrees of polymerization and molecular weights less than 3900 Da [22]. Because their low molecular weights result in low viscosity and enhanced solubility, COs have favorable biological properties that conduct numerous applications in various fields, and these applications were reviewed by Liaqat and Eltemm [23]. COs may be classified in homochitooligosaccharides containing only D-glucosamine (GlcN or D) or N-acetyl-D-glucosamine (GlcNAc or A) units and heterochitooligosaccharides, containing both GlcN and GlcNAc units. Heterooligosaccharides differ in the degree of deacetylation (the percent of GlcN units in the oligomer, DaD) and/or in the position of GlcNAc residues in the oligomer chain (acetylation pattern, AP). Both homooligosaccharides and heterooligosaccharides may differ in the number of the monomeric units within the oligomer and the degree of polymerization, DP [24]. Because their properties, heterochitooligosaccharides are preferably used in food and pharmaceutical industries [25]. Both chitosan, chitooligosaccharides and their derivatives are used in the modern agriculture as plant growth-promoting agents [26], fertilizers and biopesticides [27,28] and can be easily released into the environment.

Despite their numerous favorable pharmaceutical profiles, there are few published studies revealing predictions of low toxicological effects on humans of COs [29] and their derivatives [30]. These toxicological effects have been observed regardless of the physicochemical properties of the COs (MW, DaD and AP). A computational study emphasized favorable interactions of small COs with human plasma proteins, the interaction energies increasing with the MW, decreasing with the increase of the DaD, and being reliant on the AP [31]. Furthermore, the COs interactions with chitin deacetylases [32] and with lysozyme [33] conducted to distinct deacetylation and, respectively, degradation efficiency for the COs having dissimilar MW, DaD and AP. It underlines the necessity to consider the chemical properties of COs when analyzing their eco-toxicological effects.

The Organization for Economic Cooperation and Development (OECD) has established several standardized assays as “Guidelines for the Testing of Chemicals”. One of these assays is “Test No. 221: *Lemna* sp. Growth Inhibition Test”, which is designed to assess effects of chemicals on the freshwater aquatic macrophyte *Lemna minor* or other species of this genus [34]. This test was successfully implemented for the assessment of eco-toxicity of other polysaccharides [35].

The explosive growth in the size and diversity of data from the natural sciences and their wide accessibility were conducted for the creation of specific databases, computational packages for data manipulation and accurate computational models for computational toxicology assessment [36]. The computational methods used for environmental risk assessment have been recognized by the OECD since 2004 and described in detail since 2007 [37]. Consequently, the computational tools are used for connecting the properties of chemicals and their biological activity. They were usually designed for assessing the biological activities and/or side effects of drugs but proved to be successfully applied for other types of chemicals, such as drugs related compounds [29,30,38,39,40], artificial sweeteners and food additives [41,42], pesticides [35,42,43,44,45] and industrial chemicals [46].

Chitosan and its derivatives’ production and use in many fields may result in their release into the environment They may easily reach the aqueous environment and produce effects on the aquatic organisms. The scientific literature has proved to be poor in information regarding the possible effects of chitosan, its derivatives and COs with specific molecular properties on the aquatic organisms. The aim of this study is to assess the possible effects of these chemicals on several organisms living in aqueous environment by using both an experimental and a computational approach. The experimental approach is based on the *Lemna minor* growth inhibition assay and the computational approach uses two predictions tools to assess the possible effects of the investigated chemicals on several types of model organisms living in aqueous environments.

## 2. Results and Discussions

### 2.1. Effects of Chitosan, Chitooligosaccharides and Carboxymethyl Chitosan on Lemna minor

The number of fronds, corresponding to the exposure of *Lemna minor* to chitooligosaccharides and their derivatives (G, NAG, 2G and N-CMChi), was used to plot a dose–response curve (Figure 1a–d) and to calculate the half-maximal effective concentration of each chitooligosaccharide against *L. minor*. In this figures, symbols with same letter do not present statistically significant differences (Appendix A).

The calculated EC_50_ value against *Lemna minor* of the D-glucosamine hydrochloride (Figure 1a) corresponds to the “slightly toxic” aquatic toxicity category according to U.S. EPA [47]. This outcome is in good correlation with the results of the few studies addressing the effects of D-glucosamine on other types of duckweed. D-glucosamine determined a level of zeaxanthin and antheraxanthin approximately 30 times higher than the control in *Lemna trisulca* and induced the inhibition of violaxanthin synthesis from zeaxanthin and antheraxanthin, even at a concentration of 0.01% [48]. D-glucosamine also caused a considerable concentration-dependent membrane depolymerization in *Lemna gibba*, a 30% inhibition of the respiration rate being observed at a D-glucosamine concentrations of 50 mM [49].

To the best of our knowledge, this is the first study regarding the ecotoxicity of N-acetyl-D-glucosamine, and its calculated EC_50_ value (Figure 1b) corresponds to the “practically non-toxic” aquatic toxicity category.

A lack of knowledge was also identified regarding the ecotoxicological effects of chitobiose dihydrochloride. The calculated EC_50_ value of chitobiose dihydrochloride against *Lemna minor* (Figure 1c) corresponds to the “potentially non-toxic” category [47]. Similarly, the calculated EC_50_ value for N-CMChi (Figure 1d), revealed that N-CMChi belongs to the category of “potentially non-toxic” chemicals.

For each of the four chitosan samples, it was possible to plot a dose–response curve based on the number of fronds (Figure 2a–d). In this figures, symbols with same letter do not present statistically significant differences (Appendix A. EC_50_ values were determined for ChiS-, ChiM-, ChiL and Chi50, with all four chitosan samples being potentially non-toxic.

The only information that was identified in the specific literature regarding the effects of chitosan (DaD = 90%) on aqueous environments is the inhibition of the growth of algae such as *Chlorella vulgaris* [13].

The comparison of EC_50_ values (Figure 3) determined for tested COs and chitosan samples highlighted the fact that D-glucosamine hydrochloride had the lowest value of EC_50_ against *L. minor* and the highest value of EC_50_ was observed for CMChi. Potential non-toxic effects on *L. minor* were also observed in the cases of other polysaccharides, such as sodium alginate or carboxymethyl cellulose, with those determined EC_50_ values strongly affected by parameters such as logP and logS [35].

The scientific literature and databases lack the information regarding the EC_50_ values on *Lemna minor* for biomaterials. The only identified EC_50_ values for *L. minor* were for sodium alginate (ALG) (EC_50_ 1769.3 mg/L) and sodium carboxymethyl cellulose (CMC) (EC_50_ 2244.2 mg/L) [35]. The analyzed samples of chitosan, chitooligosaccharides and carboxymethyl chitosan exhibited lower EC_50_ in comparison with two other polysaccharides. The highest difference in EC_50_ values was observed in the case of D-glucosamine, which was 153 times lower than the value for ALG and 194 times lower than that for CMC. For the other analyzed samples, the EC_50_ values were between 1.85 and 8.75 times lower than for ALG and between 2.34 and 11.10 times lower than for CMC. 

The possible correlation between the determined EC_50_ values and parameters such as molecular weight and degree of deacetylation was analyzed (Appendix A). The molecular weight and degree of deacetylation were determined based on the SMILES formulas in the case of chitooligosaccharides derivatives and carboxymethyl chitosan, and experimentally, in the case of chitosan samples (ChiS, ChiM, ChiL and Chi50). Although no statistically significant correlation was identified between the EC_50_ values and the molecular weight corresponding to all tested samples, such correlations were still observed in the case of some samples (Appendix A). More precisely, a statistically significant correlation (*p* < 0.05) of 0.988 was identified in the case of samples 2G, ChiS, ChiM and ChiL and of −0.999 in the case of samples Chi50, ChiS and CMChi. In the case of the degree of deacetylation, no statistically significant correlation was observed with the EC_50_ values, considering all the analyzed samples, but such correlations were observed in the case of some samples (Appendix A). A correlation of −0.997 was observed in the case of the ChiS, ChiL and CMChi samples. These aspects underline the fact that both the molecular weight and the degree of deacetylation influence the toxicity of chitooligosaccharides and chitosan samples, but there may be other aspects that influence the EC_50_ values.

### 2.2. Predictions of the Effects of COs and Their Derivatives on Aquatic Organisms

The outcomes obtained using the admetSAR2.0 computational tool [50,51] and regarding the physicochemical properties and the possible toxicological effects of the investigated COs on aqueous organisms are revealed in Table 1 and Table 2. Table 1 refers only to the totally acetylated COs and to COs derivatives. Table 2 contains prediction for COs containing both neutral and ionized forms of glucosamine as it is known that this molecule may exist partially in the cation form at pH values of 5 to 9 [52].

Data presented in Table 1 and Table 2 reveal that investigated COs and their derivatives are not considered to produce toxicity against crustaceans and fish. It is widely accepted that toxicity level of chemicals against *Tetrahymena pyriformis* increases with the increasing value of pIGC_50_ [53]. In the case of the model used by admetSAR2.0 prediction tool, compounds with pIGC_50_ > −0.5 were assigned as producing *Tetrahymena pyriformis* toxicity (TPT) [50,51]. Consequently, none of the investigated compounds is considered as producing TPT. Nevertheless, the values obtained for pIGC_50_ of COs usually increase with MW and decrease with increasing DaD and, for the same MW and DaD, are reliant on AP. When comparing the neutral and cation forms of COs containing glucosamine subunits, the values of pIGC_50_ corresponding to the neutral forms are slightly lower than those corresponding to the cation forms. These results are in good agreement with known information revealing that polar compounds exhibit greater TPT as compared with the nonpolar ones [54].

The outcomes of the ADMETLab2.0 prediction tool [55,56] refer to the values of the 48 h *Tetrahymena pyriformis* IGC_50_, 96 h fathead minnow LC_50_ (LC_50_FM)) and the 48 h *Daphnia magna* LC_50_ (LC_50_DM) and are revealed in Figure 4, Figure 5 and Figure 6.

Data presented in Figure 4 reveal that totally acetylated COs emphasize the highest effects on *Tetrahymena pyriformis*. For the totally acetylated COs, the TPT increases with increasing MW, but for totally deacetylated COS, TPT decreases with the MW. In the cases of COs having similar MW and DaD, but distinct AP, there are small differences between the predicted values of the IGC_50_. These findings underline the importance of the DaD and AP in the effects of COs on *Tetrahymena pyriformis*. The COs derivatives, D-glucosamine hydrochloride (G), chitobiose dihydrochloride (2G), O-CMChi, N-CMChi and NO-CMChi reveal a slightly higher TPT than COs having higher a DaD.

Figure 5 reveals that the highest effects on fathead minnow are predicted for COs containing the bigger number of acetylated units (8A). The effects of COs against fathead minnow usually increase with the MW, decrease with increasing DaD and is reliant on the AP. The COs derivativesG, 2G, O-CMChi, N-CMChi and NO-CMChi emphasize lower effects on fathead minnow than COs. Similar qualitative predictions have also been obtained for the estimated COs effect on *Daphnia magna* (Figure 6). Bigger values of MW and lower values of DaD conducted to higher effects, and the AP also influenced the LC_50_DM values. The COs derivatives revealed lower effects on *Daphnia magna* than on COs.

All the results obtained using the computational approach are also in good agreement with known data regarding the possible human health effects produced by the investigated COs that proved to be dependent on the MW, DaD and reliant on the AP [29,31,32,33]. In order to establish which are the factors mainly affecting the ecotoxicological effects of the investigated COs and their derivatives, the predicted values for the IGC_50_, LC_50_FM and LC_50_DM obtained for totally acetylated and totally deacetylated COs were plotted taking into account the computed MW, logP and logS values. The points were fitted using a linear fitting and the equations are given in the Table 3.

The equations presented in Table 3 reveal that solubility (logS) is the property mainly affecting the effects of the investigated COs on aqueous environment, with a higher solubility conducting lower effects on the aqueous model organisms.

The scientific literature reveals a dependence of physicochemical properties and biological activity of a chemical compound on its molecular structure. The molecular structure can be described using molecular indices, and they are further used in order to compute the physicochemical properties and to predict the biological properties of chemical compounds [57,58]. One of these indices, the Wiener index, proved to be correlated with the boiling points of alkane molecules [59] and the other properties of substances such as the density, viscosity, surface tension [60] and with biological activity and/or chemical reactivity [61,62]. Within this study, the Wiener indices of the chitooligosacharides and their derivatives were computed and their possible correlation with some physicochemical properties (MW, logP, logS) and the computed ecotoxicological parameters (IGC_50_, LC_50_FM and LC_50_DM) was assessed. A statistically significant (*p* < 0.05) correlation was observed between the Wiener index and MW, logP and logS but also with IGC_50_, LC_50_FM and LC_50_DM for COs with unprotonated structures (Appendix A). For COs with protonated structures, a statistically significant correlation was observed between Wiener index and MW, logP, IGC_50_ and LC_50_FM (Appendix A). The correlation of the Wiener index with investigated ecotoxicological parameters indicates that this index can be used for predicting these biological actions. 

## 3. Materials and Methods

### 3.1. Materials Used in the Experimental Approach

D-glucosamine hydrochloride with stock keeping unit (SKU) G1514 and chitobiose dihydrochloride (SKU SMB00279) were purchased from Sigma Aldrich. N-acetyl-D-glucosamine (Order No. 8993.4), zinc chloride (Order No. 3533) and acetic acid (Order No. 3738.4) were purchased from Carl Roth. N-carboxymethyl chitosan (Catalog No. sc-358091) was purchased from Santa Cruz Biotechnology, while four Chitopharm chitosan samples with variable molecular weight and deacetylation degrees were obtained from Chitinor, in the ChitoWound project (ID 4/2017, PN3-P3-284, “Biotechnological tools implementation for new wound healing applications of byproducts from the crustacean seafood processing industry”).

Four types of polymers were tested, namely low- (ChiS), medium- (ChiM) and high- (ChiL) molecular-weight chitosan and chitosan with a degree of deacetylation of approximately 50% (Chi50). The chitosan samples were tested in three concentrations (50 mg/L, 500 mg/L and 5000 mg/L) and were obtained by dissolving the polymers in 0.3% acetic acid, followed by dilution with culture medium in a 1:2 ratio.

Four types of chitooligosaccharides were tested: D-glucosamine hydrochloride (G), N-acetyl-D-glucosamine (NAG or A), chitobiose dihydrochloride (2G), and carboxymethyl chitosan (CMChi). Three chitooligosaccharides (G, NAG, 2G) were tested in three concentrations: 5 mg/L, 50 mg/L and 250 mg/L, being dissolved directly into *L. minor* culture medium. CMChi was tested in four concentrations (50 mg/L, 250 mg/L, 500 mg/L and 5000 mg/L) and the solutions were prepared by dissolving the polymer directly in L. minor culture medium.

### 3.2. Materials Considered in the Computational Approach

Within this study we considered the COs and their derivatives presented in Table 4. COs containing up to 8 monomeric units were taken into account as they are considered as water soluble.

The Simplified Molecular Input-Line Entry System (SMILES) formulas of these COs were obtained using ACD/ChemSketch software (https://chemicalize.com–accessed on 6 July 2021) and were used as entry data for the prediction tools. For the D-glucosamine hydrochloride, chitobiose hydrochloride and the three types of CMChi, the canonical SMILES formulas were extracted from PubChem database [63].

### 3.3. Lemna minor Growth Inhibition Assay

*Lemna minor*, the common duckweed, was used as a test organism for the assessment of growth response of the tested chitosan samples and chitooligosaccharides through a growth inhibition assay. Standard conditions as described in OECD guideline [34] were maintained during both *L. minor* culture and growth inhibition assay.

The effects of the chitosan samples and the chitooligosaccharides were tested using a number of 10 fronds per test vessel, with an exposure period of 7 days. In the same conditions, two controls were tested: a negative control (C−), represented by culture medium, and a positive control (C+), represented by 0.5% zinc chloride. All samples and controls were tested in triplicate.

The endpoints of the growth inhibition test were represented by number of fronds, which was used for the plotting of dose–response curves and the calculation of half maximal effective concentration (EC_50_). The tested samples were classified into aquatic ecotoxicity categories according to U.S. EPA [47] based on the calculated EC_50_ values: very highly toxic (<0.1 mg/L), highly toxic (>0.1–1 mg/L), moderately toxic (>1–10 mg/L), slightly toxic (>10–100 mg/L) and practically non-toxic (>100 mg/L).

### 3.4. Computational Assessment of the Effects of COs on the Aquatic Organisms

In order to obtain information regarding the possible effects of the investigated COs on the aquatic organisms, the admetSAR2.0 [50,51] and ADMETLab2.0 [5,56] online prediction tools were used.

The admetSAR2.0 is a web server that, based on SMILES formula and QSAR models, allows us to compute the physicochemical properties and to estimate both ecological and mammalian absorption, distribution, metabolism, excretion and toxicity (ADMET) properties for the investigated chemical molecules. Several eco-toxicity predictions models with high accuracy are included for environmental risk assessment of chemicals and three of them correspond to aquatic toxicity and were used in the present study. There are two binary prediction models: fish aquatic toxicity (FAQ) (84% accuracy) and crustacean toxicity (CT) (77% accuracy), which determine the probabilities that investigated chemical molecules produce or not FAQ and/or CT. There also is one regression model, *Tetrahymena pyriformis* toxicity (TPT) (R2 = 0.822), which allows to estimate the TPT expressed as pIGC_50_ = −logIGC_50_ (with IGC_50_ being the concentration of a chemical that inhibits 50% of the growth of the population of *Tetrahymena pyriformis* measured in µg/L) [50,51].

ADMETLab2.0 tool also considers the SMILES formulas as entry data and, based on quantitative structure–property relationship (QSPR) models, determines with good accuracy the following information regarding the aquatic toxicity [5,56]: (i) the value of the 48 h *Tetrahymena pyriformis* IGC_50_ (the concentration of the chemical molecule in water that produces 50% growth inhibition of *Tetrahymena pyriformis* organisms after 48 h) (R^2^ = 0.86); (ii) the value of 96 h fathead minnow LC_50_ (LC_50_FM, meaning the concentration of the chemical molecule in water that causes the death of 50% of fathead minnow after 96 h, R^2^ = 0.66); and (iii) the value of 48 h *Daphnia magna* LC_50_ (LC_50_DM, meaning the concentration of the chemical molecule in water that causes the death of 50% of *Daphnia magna* after 48 h, R^2^ = 0.91). All these predictions are often used to evaluate the aquatic toxicity endpoints. Furthermore, ADMETLab2.0 tool has been considered for computing the logarithm of aqueous solubility (logS) expressed in log mol/L using a regression model with R^2^ = 0.871. 

Furthermore, a molecular indices analysis was considered based on computing the Wiener index for the investigated chitooligosaccharides and their derivatives. The Wiener index, the most widely used distance based topological index, is defined as the sum of the chemical distances between all the pairs of vertices in a molecular graph representing the nonhydrogen atoms in a molecule [59]. The Wiener indices for the investigated COs and their derivatives were computed using the Wiener Index Calculator online software (Supercomputing Facility for Bioinformatics & Computational Biology, Indian Institute of Technology, New Delhi, India, available online http://www.scfbio-iitd.res.in/software/drugdesign/wienerindex.jsp–accessed on 7 September 2022). The entry data were the 3D pdb files of the COs and their derivatives that were built from their SMILES formulas using the Online SMILES Translator and Structure File Generator (Computer-Aided Drug Design Group of the Chemical Biology Laboratory, Bethesda, MD, USA, available online https://cactus.nci.nih.gov/translate/–accessed on 7 September 2022). The possible correlation between the computed Wiener indices and the physicochemical properties and predicted ecotoxicological parameters was analyzed. 

### 3.5. Statistical Analysis Used in the Experimental Approach

PAST software [64] was used for the statistical analysis of data, with Quest Graph™ EC_50_ Calculator [65] being used for the calculation of EC_50_ values. The Shapiro–Wilk W test was used for assessing the normality of the data, with the distribution analysis being followed by an ANOVA analysis. The normally distributed data were analyzed with parametric tests, the homogeneity of variance among treatments was determined through Levene’s test, with Tuckey’s post hoc test being used for the analysis of variances. The non-normally distributed data were analyzed with the Kruskal–Wallis test, variances being further analyzed using Dunn’s post hoc test. The Pearson linear r correlation test was used for all correlation analysis. The differences were considered significant for *p* values < 0.05.

## 4. Conclusions

The novelty of this study consists in experimental and computational assessment of several ecotoxicological data for chitosan with variable molecular weights and deacetylation degrees, chitooligosaccharides and their derivatives, i.e., the EC_50_ values against *Lemna minor*, the 48 h *Tetrahymena pyriformis* IGC_50_ values, the 96 h fathead minnow LC_50_ values and the 48 h *Daphnia magna* LC_50_ values, respectively. The experimentally determined EC_50_ values against *Lemna minor* revealed that D-glucosamine hydrochloride is the only compound that is considered as “slightly toxic” for aquatic environment (EC_50_ = 11.55 mg/L), with all the other investigated molecules being considered as “practically non-toxic” (EC_50_ > 100 mg/L). These outcomes were in very good correlation with the results of the computational approach revealing that investigated chitooligosaccharides and their derivatives were not considered to produce toxicity against crustacean, fish and *Tetrahymena pyriformis* microorganisms. The lowest values of IGC_50_ for *Tetrahymena pyriformis*, LC_50_FM and LC_50_DM are revealed by totally acetylated COs. Data obtained both experimentally and computationally revealed that molecular weight, the degree of deacetylation and the acetylation pattern influenced the effects of the chitooligosaccharides and chitosan on *aquatic organisms*. In the case of chitooligosaccharides, the solubility was the property mainly influencing the effects on aqueous environment, a higher solubility resulted in lower negative effects. As the release into the environment of chitosan, chitooligosaccharides and their derivatives may result from their production and/or use in numerous fields, it becomes necessary to investigate their possible effects on the aqueous environment and to assess how their characteristics are reflected in their potential toxicity.

## Figures and Tables

**Figure 1 molecules-27-06123-f001:**
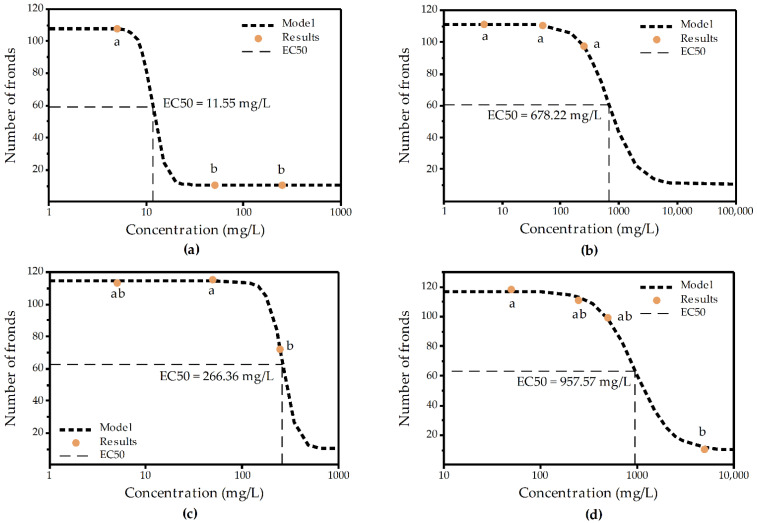
Dose–response curve of the number of fronds of *Lemna minor* and the concentrations of (**a**) D-glucosamine hydrochloride (G), (**b**) N-acetyl-D-glucosamine (NAG), (**c**) chitobiose dihydrochloride (2G) and (**d**) N-carboxymethyl chitosan (N-CMChi). Symbols with same letter do not present statistically significant differences (Appendix A).

**Figure 2 molecules-27-06123-f002:**
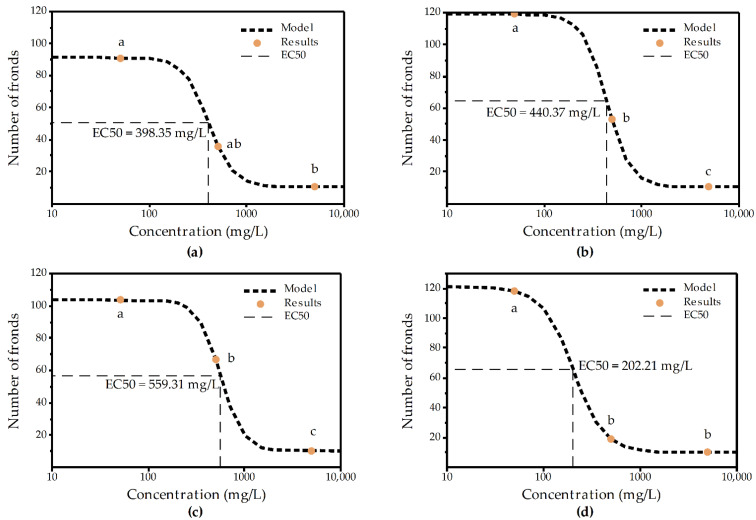
Dose–response curve of the number of fronds and the concentrations of (**a**) chitosan with low molecular weight (ChiS), (**b**) chitosan with medium molecular weight (ChiM), (**c**) chitosan with high molecular weight (ChiL) and (**d**) chitosan with a deacetylation degree of approximately 50% (Chi50). Symbols with same letter do not present statistically significant differences (Appendix A).

**Figure 3 molecules-27-06123-f003:**
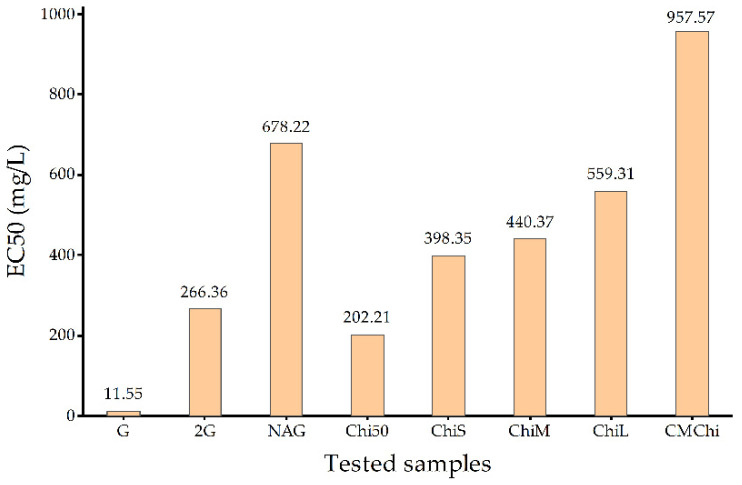
Comparison of EC_50_ values of tested chitooligosaccharides, chitosan and their derivatives.

**Figure 4 molecules-27-06123-f004:**
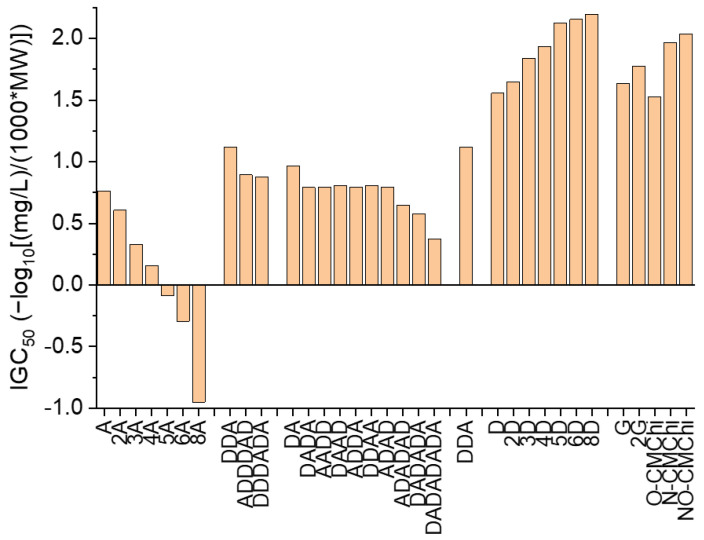
48 h *Tetrahymena pyriformis* IGC_50_ predicted values for chitooligosaccharides and their derivatives using ADMETLab2.0. The meaning of acronyms on the OX axe are explained in Table in Section 3.2.

**Figure 5 molecules-27-06123-f005:**
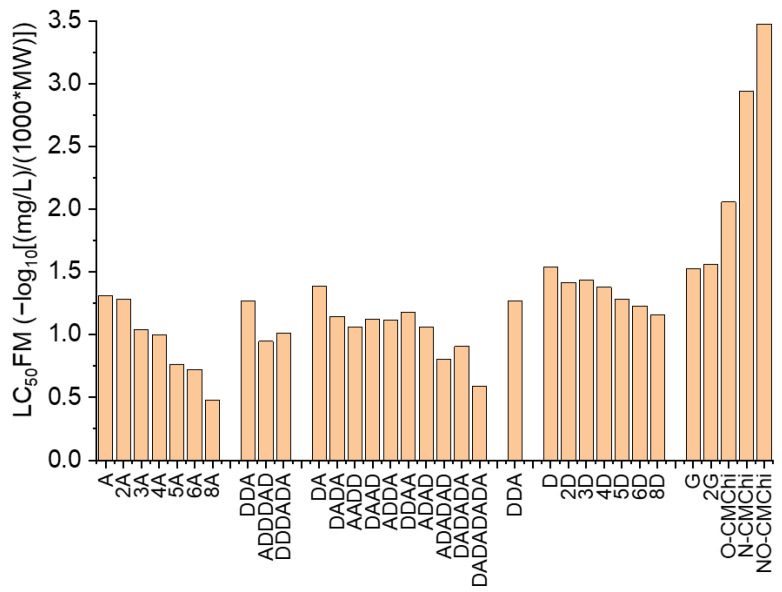
The 96 h fathead minnow LC_50_ predicted values for chitooligosaccharides and their derivatives using ADMETLab2.0. The meaning of acronyms on the OX axe is explained in Table in Section 3.2.

**Figure 6 molecules-27-06123-f006:**
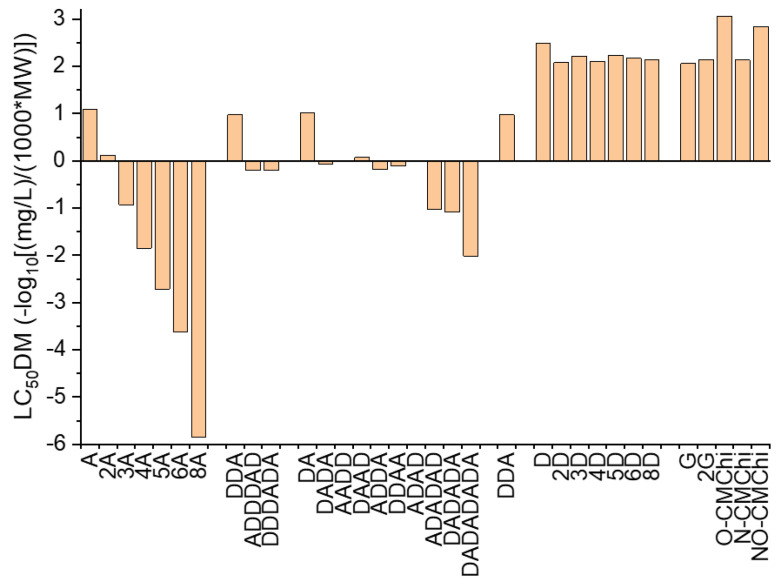
48 h *Daphnia magna* LC_50_ predicted values for chitooligosaccharides and their derivatives using ADMETLab2.0. The meaning of acronyms on the OX axe are explained in Table in Section 3.2.

**Table 1 molecules-27-06123-t001:** Physicochemical properties and predictions obtained using admetSAR2.0 tool and regarding the effects of totally acetylated chitooligosaccharides (COs) and of COs derivatives on aquatic organisms: MW—molecular weight, logP—partition coefficient, pIGC_50_ = −logIGC_50_ (IGC_50_ is the concentration of a chemical that inhibits 50% of the growth of the population of *Tetrahymena pyriformis* measured in µg/L). The meaning of the acronyms is explained in Table in Section 3.2.

COs/COs Derivatives	MW (g/mol)	LogP	Probability to Produce		pIGC50 (µg/L)
			Crustacea Toxicity	Fish Aquatic Toxicity	
A	179.170	−2.143	−0.790	−0.959	−1.446
2A	340.330	−3.032	−0.710	−0.875	−1.066
3A	501.480	−3.658	−0.710	−0.865	−0.959
4A	662.640	−4.545	−0.710	−0.865	−0.766
5A	823.790	−5.178	−0.710	−0.865	−0.738
6A	984.950	−6.099	−0.710	−0.865	−0.730
8A	1037.260	−7.713	−0.710	−0.865	−0.652
D-glucosamine hydrochloride	170.080	−0.073	−0.900	−0.981	−1.341
chitobiose dihydrochloride	340.150	0.212	−0.880	−0.977	−0.963
O-CMChi	273.210	−3.150	−0.760	−0.971	−1.226
N-CMChi	295.240	−3.470	−0.740	−0.959	−0.778
NO-CMChi	353.280	−3.630	−0.705	−0.922	−0.806

**Table 2 molecules-27-06123-t002:** Predictions obtained using admetSAR2.0 tool and regarding the physicochemical properties and the effects of partially deacetylated and totally deacetylated chitooligosaccharides (COs) in both neutral and cation forms: MW—molecular weight, logP—partition coefficient, pIGC_50_ = −logIGC_50_ (IGC_50_ is the concentration of a chemical that inhibits 50% of the growth of the population of *Tetrahymena pyriformis* measured in µg/L). The meaning of acronyms is explained in Table in Section 3.2.

DaD	COs	MW (g/mol)	LogP	Cation Form			Neutral Form		
				Probability to Produce		pIGC_50_ (µg/L)	Probability to Produce		pIGC_50_ (µg/L)
				Crustacea Toxicity	Fish Aquatic Toxicity		Crustacea Toxicity	Fish Aquatic Toxicity	
33%	ADA	585.560	−3.332	−0.690	−0.917	−1.000	−0.700	0.871	−0.925
50%	DA	382.360	−2.807	−0.690	−0.954	−1.239	−0.690	−0.929	−1.253
	DADA	746.710	−4.119	−0.690	−0.917	−0.959	−0.680	−0.871	−0.922
	AADD	746.710	−4.119	−0.690	−0.917	−0.921	−0.700	−0.871	−0.862
	DAAD	746.710	−4.073	−0.690	−0.917	−0.798	−0.680	−0.871	−0.838
	ADDA	746.710	−4.194	−0.690	−0.917	−0.871	−0.700	−0.871	−0.846
	ADADAD	746.710	−4.138	−0.690	−0.909	−0.700	−0.700	−0.861	−0.684
	DADADA	746.710	−4.119	−0.690	−0.909	−0.906	−0.680	−0.861	−0.890
	DADADADA	1857.770	−6.858	−0.690	−0.909	−0.894	−0.680	−0.861	−0.862
67%	DDA	523.520	−3.664	−0.690	−0.954	−1.015	−0.680	−0.929	−0.969
	ADDDAD	1069.020	−5.772	−0.690	−0.917	−0.650	−0.700	−0.871	−0.664
	DDDADA	1069.020	−5.737	−0.690	−0.917	−0.904	−0.680	−0.871	−0.895
100%	D	179.170	−2.143	−0.860	−0.979	−1.481	−0.870	−0.966	−1.283
	2D	340.330	−3.032	−0.790	−0.963	−1.210	−0.770	−0.941	−1.058
	3D	501.480	−3.658	−0.790	−0.963	−0.885	−0.770	−0.941	−0.909
	4D	662.640	−4.545	−0.790	−0.963	−0.884	−0.770	−0.941	−0.866
	5D	823.790	−5.178	−0.790	−0.963	−0.821	−0.770	−0.9416	−0.829
	6D	984.950	−6.099	−0.790	−0.963	−0.848	−0.770	−0.9416	−0.830
	8D	1037.260	−7.713	−0.790	−0.963	−0.810	−0.770	−0.9416	−0.807

**Table 3 molecules-27-06123-t003:** Dependence of the predicted values for the aqueous toxicity of investigated chitooligosaccharides on their physicochemical properties: MW—molecular weight, logS—aqueous solubility, logP—partition coefficient, DaD—deacetylation degree.

Physicochemical Property/Deacetylation Degree	DaD = 0%		DaD = 100%	
	Equation	R^2^	Equation	R^2^
MW	IGC50 = −0.0002 MW + 1.099	0.985	IGC50 = −0.0007 MW + 1.420	0.978
	LC50FM = −0.0006 MW + 1.473	0.967	LC50FM = −0.0004 MW + 1.602	0.936
	LC50DM = −0.0048 MW +2.188	0.997	LC50DM = −0.0002 MW + 2.335	0.242
logS	IGC50 = −1.040 logS + 0.350	0.978	IGC50 = 0.348 logS + 1.700	0.833
	LC50FM = −0.707 logS + 1.083	0.909	LC50FM = −0.186 logS + 1.462	0.888
	LC50DM = −5.580 log S − 0.850	0.956	LC50DM = −0.073 logS + 2.245	0.229
logP	IGC50 = 0.43 logP + 1.640	0.994	IGC50 = −0.181 logP + 1.347	0.887
	LC50FM = 0.22 logP + 1.741	0.956	LC50FM = 0.030 logP + 2.362	0.991
	LC50DM = 1.750 logP + 4.340	0.998	LC50DM = 0.067 logP + 1.664	0.998

**Table 4 molecules-27-06123-t004:** Chitooligosaccharides and their derivatives considered in the computational study (A—unit of N-acetyl-glucosamine, D—unit of glucosamine, DaD—deacetylation degree, G—D-glucosamine hydrochloride, 2G—chitobiose hydrochloride, O-CMChi—O-carboxymethyl chitosan, N-CMChi—N-carboxymethyl chitosan, NO-CMChi—N-O-carboxymethyl chitosan.

Homo-Chitooligosaccharides		Hetero-Chitooligomers			Derivatives
AP = 0	AP = 100%	AP = 33%	AP = 50%	AP = 67%	
2D, 3D, 4D, 5D, 6D, 8D	2A, 3A, 4A, 5A, 6A, 8A	DDA, DDDADA ADDDAD	DA, AADD, ADAD, ADDA, DAAD, DDAA, DADA, ADADAD, DADADA, DADADADA	ADA	G, 2G, O-CMChi, N-CMChi, NO-CMChi

## Data Availability

The data presented in this study are available in Appendix A.

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
