# Peer review of "Assessment of the Effects of Chitosan, Chitooligosaccharides and Their Derivatives on *Lemna minor"

_molecules, 2022, doi:10.3390/molecules27186123_

Round 1
Reviewer 1 Report
The selected research topic is really relevant and interesting. the study is very well conducted and good results have been obtained. Regarding the presentation of work, the introduction is too exhaustive and can be shortened with more clear language. The results and discussion part presented in can be simplified. The data which are presented in the figure or tables may be removed from the text and the results may be presented in simpler words. Language corrections are required throughout the manuscript.
Author Response
Thank you very much for paying attention to our study and for your valuable suggestions that are clearly meant to improve the quality of the manuscript. Your suggestions are presented in italics and our answer is normal text. Also, in the text of the manuscript, the changes that are done taking into account your observations are made using “track changes” and explained in the following.
- The selected research topic is really relevant and interesting. the study is very well conducted and good results have been obtained. Regarding the presentation of work, the introduction is too exhaustive and can be shortened with more clear language.
The introduction has been shortened, there are sentences that are removed of shortened, as you may notice in the revised form of the manuscript (lines 25-141).
- The results and discussion part presented in can be simplified.
We simplified as much as possible the results and discussion part (lines 153- 361). However, some new information has been added as requested by Reviewer 2.
- The data which are presented in the figure or tables may be removed from the text and the results may be presented in simpler words.
The data that are present in tables and / or figures have been removed from the text. The Results section has been modified.
- Language corrections are required throughout the manuscript.
We did our best to correct the language within the entire manuscript.

Reviewer 2 Report
Comments to the Author
The article describes the Assessment of the effects of chitosan, chitooligosaccharides and their derivatives on Lemna minor. However, there are some issues need to be fixed.
1. More specific data must be mentioned in abstract
2. What is the novelty of this study? Since a number of papers have been published on chitosan, and also chitooligosaccharides, the authors should clearly specify what is new?
3. I suggest authors to add some molecular indexes analysis so as to increase publication value of the paper.
4. The paper lacks molecular mechanism. In my opinions, results lack deep insight.
5. I think the properties of the Chitosan, chitooligosaccharides and their derivatives should be compared with other commercial mateials reported previously in main text, such as active materials and biopolymers. Otherwise, it difficult to claim the superiority of this chitosan-based wound dressing.
Author Response
Thank you very much for paying attention to our study and for your valuable suggestions that are clearly meant to improve the quality of the manuscript. Your suggestions are presented in italics and our answer is in normal text. Also, in the text of the manuscript, the changes that are done taking into account your observations are made using “track changes” and explained in the following.
- More specific data must be mentioned in abstract.
The abstract has been organized taking into account the Instructions for Authors and must specifies the background, the used methods, the results and conclusions in 200 words. We made changes taking into account your observation, but also fulfilling the Instructions for Authors.
- What is the novelty of this study? Since a number of papers have been published on chitosan, and also chitooligosaccharides, the authors should clearly specify what is new?
There are numerous published papers on chitosan and chitooligosacharides, but they address mainly the pharmaceutical effects, not the ecotoxicological effects. We were not able to find in specific databases and scientific literature EC50 and LC50 values for these biomolecules for the aquatic organisms. A paragraph has been added at the beginning of the Conclusions section to explain the novelty of our study (lines 472-476).
- I suggest authors to add some molecular indexes analysis so as to increase publication value of the paper.
We computed of the Wiener indices and correlated them with both the physicochemical properties and ecotoxicological parameters of the investigated molecules. A new paragraph has been added in the Results section (lines 344-359), two supplementary tables (S9 and S10) have been added in Supplementary information file and a new paragraph in the Methodology section such as to describe the computation of the Wiener indices (lines 444-458). New references have been also added to support the new information.
- The paper lacks molecular mechanism. In my opinions, results lack deep insight.
Within this study, there are numerous compounds considered for assessing some of their ecotoxicological effects. The aim of our study was not the elucidation of the molecular mechanisms of the toxicity of these molecules. Elucidating the molecular mechanism of the biological/toxicological for the effect of chemicals is not a trivial task. It involves numerous experimental analysis and could be the subject of another study.
- I think the properties of the Chitosan, chitooligosaccharides and their derivatives should be compared with other commercial materials reported previously in main text, such as active materials and biopolymers. Otherwise, it difficult to claim the superiority of this chitosan-based wound dressing.
Our manuscript does not deal with the wound healing properties of chitosan, chitooligosaccharides and / or their derivatives. Even if these biomaterials have wound healing properties, they are only mentioned in the manuscript as applications of these compounds. However, we have compared thevalues of the EC50 on Lemna minor obtained for chitosan and chitooligosacharides with published data regarding the EC50 values obtained for the same organisms but for other biopolymers: alginate and sodium carboxymethyl cellulose. Consequently, a new paragraph has been introduced (lines 204-212).
Round 2
Reviewer 2 Report
Accept